

# Diurnal pattern of salivary alpha-amylase and cortisol under citric acid stimulation in young adults

Haimei Xie[1,2,*], Xiaomei Zheng[1,3,*], Ye Huang[1], Weihao Li[1], Wenkai Wang[1], Qiao Li[3], Jiangtao Hou[3], Lulu Luo[2], Xiuying Kuang[3] and Chuan-quan Lin[1]

[1] Guangzhou University of Traditional Chinese Medicine, Science and Technology Innovation Center, Guangzhou, Guangdong, China
[2] Guangzhou University of Traditional Chinese Medicine, Guangdong Provincial Hospital of Chinese Medicine, Guangzhou, Guangdong, China
[3] Guangzhou University of Traditional Chinese Medicine, First Affiliated Hospital of Guangzhou University of Chinese Medicine, Guangzhou, Guangdong, China
[*] These authors contributed equally to this work.

Corresponding author
Chuan-quan Lin,
chuanquanlin@gzucm.edu.cn

## ABSTRACT

**Background**. Saliva composition has diurnal variations. Citric acid stimulation plays a major role in the change of salivary flow rate and salivary composition. However, diurnal variations and sex differences in salivary alpha-amylase (sAA), pH, salivary flow rate (SFR), and salivary cortisol before and after citric acid stimulation remain unclear.

**Methods**. We recruited 30 healthy volunteers, including 15 women ($24.7 \pm 1.0$ years old) and 15 men ($25.3 \pm 1.3$ years old). At four time points (T1, 7:00; T2, 10:00; T3, 16:00; and T4, 20:00), saliva was collected from healthy volunteers before and after citric acid stimulation; and sAA, pH, SFR and salivary cortisol were measured and compared between men and women.

**Results**. There were circadian fluctuations in sAA activity, SFR, pH, and cortisol level both before and after citric acid stimulation, and the diurnal fluctuations of these indexes were not affected by citric acid stimulation. There were significant differences in salivary cortisol between men and women before and after acid stimulation in T1. Neither SFR nor pH showed sex-related differences before or after acid stimulation. The variation trend of sAA activity was contrary to that of cortisol, with a significant negative correlation.

**Conclusions**. Our data suggest that sAA and cortisol showed diurnal fluctuation, and the variation characteristics of male and female under resting state and acid stimulation were basically the same. The variation trend of salivary alpha-amylase activity was opposite to that of cortisol, with significant negative correlation. Our findings may enable the selection of the correct sampling time for research and the selection of appropriate sampling strategies in studies investigating chronic psychosocial conditions.

## INTRODUCTION

Circadian rhythm is a universal living phenomenon that plays an important role in maintaining various physiological functions of plants and animals and in regulating their adaptability to internal and external environments (*Huang et al., 2020*). Various physiological indicators of the human body, such as heart rate, blood pressure, urine volume, and body temperature, fluctuate in a circadian rhythm (*Bollinger & Schibler, 2014*). Salivary secretion also varies in a circadian manner. Salivary amylase (sAA) is the main protein (about 40%–50%) in saliva (*Oppenheim et al., 2007*; *Humphrey & Williamson, 2001*; *Noble, 2000*) and sAA activity is often a primary indicator of salivary secretion. sAA shows a distinct diurnal contour pattern, with a significant decrease within 1 hour from awakening and a steady increase in activity throughout the day (*Nater et al., 2007*). In addition, salivary cortisol levels also show diurnal variation (*Nater et al., 2007*; *Kirschbaum & Hellhammer, 1989*). In stress test studies, sAA and cortisol are considered the main indicators of the sympathetic nervous system (SNS) and hypothalamic-pituitary-adrenal axis (HPA axis), respectively (*Nater et al., 2007*). Therefore, it is important to obtain the diurnal patterns of these two indicators and their determinants. This information will enable researchers to select the right timing for acute stress studies and appropriate sampling strategies for studies investigating chronic psychosocial conditions.

Saliva is mostly secreted by three salivary glands: the parotid gland, submandibular gland, and sublingual gland. The secretion mode and the composition of unstimulated whole saliva differ significantly depending on the stimulus. In the resting state, saliva is mainly secreted by the submandibular gland, and it contains less sAA and more mucus. In the stimulated state, saliva is mainly secreted by the parotid gland, and sAA content increases and mucus decreases significantly (*Humphrey & Williamson, 2001*; *Noble, 2000*; *Gröschl & Rauh, 2006*). Most studies have focused on the circadian rhythm of mixed saliva in the resting state and less on the circadian characteristics of mixed saliva in the stimulated state (*Rohleder & Nater, 2009*). In particular, it is not clear whether circadian rhythms exist in the salivary glands' ability to respond to stimulation by citric acid—the most common form of taste stimulation of the salivary glands. Currently, most individuals have five basic tastes (sweet, sour, salty, bitter and umami) (*Dulac, 2000*; *Niki et al., 2010*). Sour taste, followed by salt (NaCl), sweet, and bitter, is the most irritating to oral mucosa and easily leads to an increase in saliva flow rate (*Dawes & Watanabe, 1987*). Constant taste stimulation usually results in varying degrees of adaptation, with sweet flavors being the most adaptive and sour flavors the least (*Matsuo, 2000*). In the case of various external stimuli (citric acid stimulation), the content of sAA will increase with the increase of parotid gland secretion (*Stroud et al., 2009*; *Schoofs, Preuss & Wolf, 2008*). Gustatory stimulation, especially acid stimulation, plays a major role in elevated SFR and altered salivary composition (*Gröschl & Rauh, 2006*; *Lee & Wong, 2009*; *Chen et al., 2015*; *De Almeida et al., 2008*). SFR and pH may affect by citric acid stimulation. Mental stress also can affect SFR, where a passionate emotional lead to increased SFR, and an uneasy emotional state lead to decreased SFR (*Bakke et al., 2004*; *Gemba, Teranaka & Takemura, 1996*). Under normal circumstances, bicarbonate and phosphate contained in saliva have strong acid-base buffer capacity, which

can make the fluctuation of saliva pH within the normal range (*Cohen & Khalaila, 2014*). Saliva pH value will affect the activity of sAA, and then affect the taste of food, food intake and digestion. Saliva flow rate is the secretion rate of saliva per unit time, which is an important index reflecting the secretion function of saliva, and an important objective index in the diagnosis of dry mouth (*Yu, Liu & Fang, 2021*). In addition, the relationship between the two classic indicators of emotional stress (sAA activity and salivary cortisol) has not been well established; especially, the relationship between the two before and after acid load has not been reported. Citric acid stimulation can directly change the secretion of saliva by the salivary gland (such as SFR, pH, sAA and cortisol levels, *etc.*), the process can be further found that the secretion of the salivary glands than the resting state of saliva biochemical information and physiological characteristics of the secretion of saliva, can also further understand the characteristics of diurnal fluctuation of saliva secretion. Furthermore, the ability of salivary glands to respond to acid loading is an important manifestation of salivary gland function.

Salivary secretion is also influenced by gender, and many hormonal systems show gender-related differences in basal activity or stress response patterns, such as HPA axis and SNS (*Kirschbaum et al., 1999*). Women seem to have a weaker response to the stimulation-activating norepinephrine (*Da Mata et al., 2020*), while catecholamines levels vary with the menstrual cycle (*Kirschbaum et al., 1999*). Therefore, it is expected that basal sAA activity and stress response would also be influenced by gender or sex hormones. However, many studies on psychological stress stimuli have found no gender differences in sAA activity between resting and stimulated states (*Nater et al., 2007*; *Gustafson & Kalkhoff, 1982*; *Goldstein, Levinson & Keiser, 1983*). Hence, it should be further verified whether sex influences salivary secretion, and specifically, whether there are gender differences in the circadian rhythm of salivary secretion under acid loading.

We, therefore, carried out the current study to identify the diurnal patterns in sAA, SFR, pH, and salivary cortisol before and after citric acid stimulation, in addition to sex differences in these indexes.

## MATERIALS & METHODS

### Participants

The study was carried out at the First Affiliated Hospital of Guangzhou University of Chinese Medicine in July 2020. We recruited 30 healthy subjects, including 15 women (24.7 ± 1.0 years old) and 15 men (25.3 ± 1.3 years old). Exclusion criteria included autoimmune diseases, musculoskeletal diseases, infectious diseases, malignant diseases, recent surgery or trauma, any oral diseases (such as periodontitis, etc.), smoking in the previous 3 months, drinking alcohol in the previous week, etc. The women who taking chemical contraceptives and a specific menstrual cycle phase were also excluded in our study. And all the exclusion criteria were objectively evaluated by three authors (Chuanquan Lin, Qiao Li and Jiangtao Hou). The study was conducted in accordance with the guidelines laid down in the Declaration of Helsinki, and all procedures involving human volunteers were approved by the Ethical review approval from the First Affiliated Hospital

of Guangzhou University of Chinese Medicine (No.K [2019] 151). All the participants provided written informed consent.

## Saliva collection methods

To minimize possible confounding effects of circadian rhythms in sAA and SFR, saliva sampling in participants was carried out at 7:00 (T1), 10:00 (T2), 16:00 (T3), and 20:00 (T4) in a bright quiet room after resting comfortably seated for 15 min. The participants refrained from eating, exercising, using toothpaste, chewing gum, or drinking any beverages one hour before sampling. All of data collection occurred in the lab. The participants went back to their student dormitories and followed their daily routine without any exercise between sample collections. The rested and acid-stimulated samples were collected by rotating a Salivette® swab (Sarstedt), without chewing or biting the swab, at a steady rotation speed (7 times per min) for 3 min before and after acid stimulus. The swab was then centrifuged (3500 g, 10 min, 4 °C) to obtain the saliva sample. The specific operation methods and details have been described in detail in our previous research (*Li-Hui et al., 2016*). After sampling, saliva was centrifuged at 5000 g at 4 °C for 15 min. The supernatant from saliva was collected and stored at − 80 °C for subsequent measurements of sAA activity and salivary cortisol.

## Determination of salivary indicators

The SFR, defined as ml/min, was measured immediately after collection using the gravimetric method (*Jensen & Vissink, 2014*). Salivary pH was measured immediately after saliva collection using a laboratory pH meter (FE20, METTLER TOLEDO Ltd., Switzerland). The sAA activity was determined using an enzymatic hydrolysis assay of the chromogenic substrate maltose (*Bernfeld, 1955*) with absorbance values detected by ultraviolet spectrophotometer at 540 nm. The specific operation methods and details have been described in detail in our previous research (*Li-Hui et al., 2016*). The cortisol test with a Roche automatic electrochemiluminescence immunoassay system Cobas E411 analyzer from Roche Diagnostics GmbH 20162402565 of Elecsys Cortisol II kit was used for testing.

The sAA, SFR, pH, and salivary cortisol levels before and after acid loading were analyzed, and the salivary index ratio (after acid stimulation/before acid stimulation) was used to reflect the salivary response to stress.

## Statistical analysis

After ascertaining by Schapiro–Wilk test that the outcome variables were not distributed normally, the values measured at the four time points were mutually compared by Friedman test. Then, post-hoc pairwise comparisons by Wilcoxon signed-rank test were used. Furthermore, the differences in the outcome variables at each time point between men and women were analyzed by Mann-Whitney *U* test. All the presented data are expressed as median. Pearson was used to analysis the correlation sAA and cortisol. Analyses were performed with SPSS 18.0 The significance level was set at $P < 0.05$. Two-sided significance tests were used throughout.

## RESULTS

### sAA, SFR, pH, and salivary cortisol at four time points before and after citric acid stimulation

#### sAA, SFR, pH, and salivary cortisol before citric acid stimulation

An analysis of the indexes at the four time points (T1–T4) revealed that sAA activity increased from morning to evening ($P < 0.01$, Fig. 1A); there were significant differences between the time points, except between T3 and T2, and between T3 and T4 ($P < 0.01$, Fig. 1A). Overall, cortisol showed a decrease from T1 to T4 ($P < 0.01$, Fig. 1B. The pH levels at T3 and T4 were significantly higher than those at T1 or T2, but there were no significant differences between T1 and T2, and between T3 and T4 (Fig. 1C). The SFR did not significantly fluctuate among the four time points (Fig. 1D).

#### sAA, SFR, pH, and salivary cortisol after citric acid stimulation

After acid stimulation, the trends in saliva sAA activity, cortisol level, and flow rate among the four time points (T1–T4) were mostly consistent with those before citric acid stimulation (Figs. 1A, 1B, 1D). Similar to the trend of sAA activity, the pH value presented a trend to increase over time (Fig. 1C), and after citric acid stimulation, the pH value at T4 was significantly higher than that at T1 and T2 (Fig. 1C). Before citric acid stimulation, sAA, pH and SFR were higher than acid stimulation. sAA in T3 and T4 was significantly higher than acid stimulation ($P < 0.01$, $P < 0.05$, Fig. 1A), pH in T1 was higher than acid stimulation ($P < 0.01$, Fig. 1C), SFR in T2 and T3 was significantly higher than acid stimulation ($P < 0.05$, $P < 0.05$, Fig. 1D) , while salivary cortisol in T1 was significantly lower than acid stimulation ($P < 0.05$, Fig. 1B).

### Gender differences in sAA activity, cortisol, SFR, and pH levels before and after acid stimulation

*Gender differences in sAA activity, cortisol, SFR, and pH before citric acid stimulation* The trend of fluctuations in sAA activity and SFR levels among the four time points was consistent, and there was no significant difference between men and women (Figs. 2A, 2D). sAA activity in both male and female groups showed a trend to increase over time, with activity at T1 significantly lower than that at three other time points ($P < 0.05$, $P < 0.05$, $P < 0.05$, Fig. 2A). SFR only in female volunteers was significantly lower at T1 than at T2 and T4 (Fig. 2D). The trend of decreasing cortisol levels over time in both men and women was generally consistent; women had a significantly higher cortisol level at T1 than men ($P < 0.01$, Fig. 2B). According to Fig. 2C, the increase in pH over time was more pronounced in women than in men, with women showing a significantly lower pH at T1 than men ($P < 0.05$), and men showing less fluctuations among the four time points.

*Gender differences in sAA activity, cortisol, SFR, and pH after citric acid stimulation.* The results showed the consistency of sAA activity and SFR fluctuations between men and women among the four time points, and no significant differences between the two groups (Figs. 2A, 2D). The fluctuations of cortisol in men and women were consistent with those before citric acid stimulation; however, in women, cortisol levels declined more significantly over time and were significantly higher than those in men at T1 ($P < 0.01$, Fig. 2B). Similar

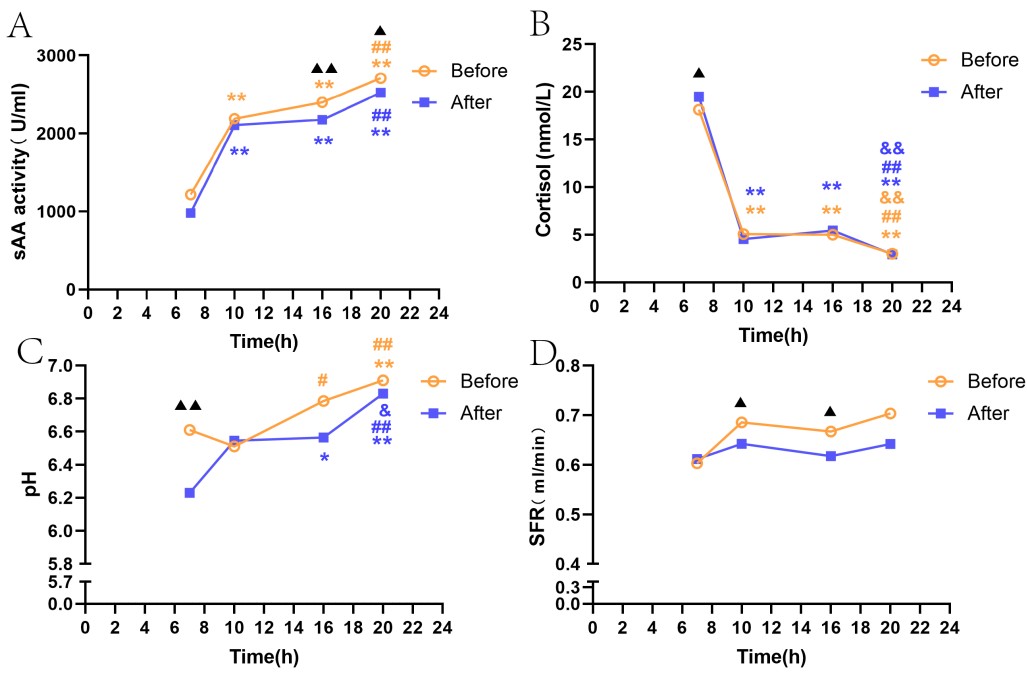

**Figure 1  SAA, SFR, pH, and salivary cortisol at four time points before and after citric acid stimulation.** Note: * $P < 0.05$, ** $P < 0.01$, comparison with T1. # $P < 0.05$, ## $P < 0.01$, comparison with T2. & $P < 0.05$, && $P < 0.01$, comparison with T3. ▲ $P < 0.05$, ▲▲ $P < 0.01$, resting state and acid stimulation comparison. Before: resting state. After: after citric acid stimulation. (A) sAA activity. (B) Cortisol. (C) pH. (D) SFR.

to the results before citric acid stimulation, women's pH fluctuated more significantly over time, while the fluctuations among the four time points were not apparent in the male group (Fig. 2C).

*Correlation between sAA activity and cortisol before and after acid stimulation.* Figure 3 show that sAA activity and cortisol levels in both groups of healthy volunteers showed opposite trends among the four time points both before and after acid stimulation, with a significant negative correlation (before acid stimulation: $r = -0.434$, $P = 0.001$, after acid stimulation: $r = -0.423$, $P = 0.001$).

## DISCUSSION

Secreted components of total saliva are usually directly related to the state of stimulation of the salivary glands, and salivary secretion is characterized by circadian rhythm (*Dawes, 1972*; *Dawes, 1975*; *Nater et al., 2007*; *Huang et al., 2020*). sAA and cortisol, biomarkers of sympathetic and HPA axis activity, have been widely investigated for their variation characteristics and circadian rhythms under psychological stress tests (*Jenzano, Brown & Mauriello, 1987*; *Rantonen & Meurman, 2000*). Gustatory stimulation is a common way to stimulate salivary glands, which is of great significance to evaluate their secretory function. However, few studies have focused on the circadian characteristics of whole saliva and

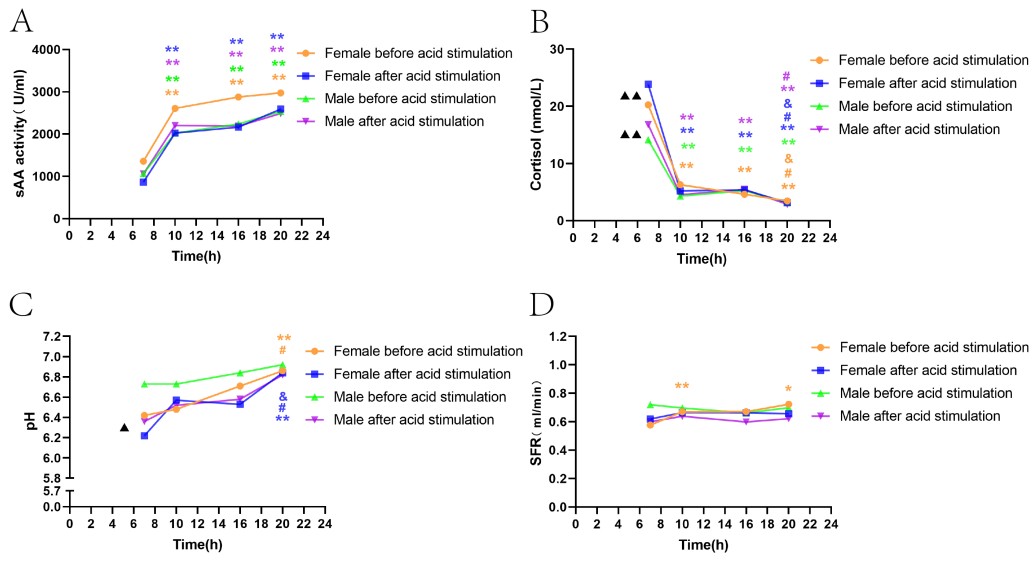

**Figure 2** **Gender differences in sAA activity, cortisol, SFR, and pH levels before and after acid stimulation.** Note: * $P < 0.05$, ** $P < 0.01$, comparison with T1. # $P < 0.05$, ## $P < 0.01$, comparison with T2. & $P < 0.05$, && $P < 0.01$, comparison with T3. ▲ $P < 0.05$, ▲▲ $P < 0.01$, gender comparison. (A) sAA activity. (B) Cortisol. (C) pH. (D) SFR.

on the circadian rhythms of sAA and cortisol during gustatory stimulation (*Rohleder & Nater, 2009*). sAA has been proposed as a marker of the activity of the SNS (*Nater et al., 2007*); sAA is produced locally in the salivary glands, controlled by the autonomic nervous system. The diurnal variation patterns and gender differences of sAA and cortisol before and after acid stimulation are still rarely reported. However, our study found that sAA, cortisol, pH and SFR all had diurnal fluctuations, among which sAA, cortisol and pH fluctuated significantly. Consistent with our earlier findings, salivary cortisol levels at T1 were significantly higher in women than in men before and after acid stimulation, and pH levels were significantly lower in women than in men (*Prodan et al., 2015*; *Li-Hui et al., 2016*).

To the best of our knowledge, the current study is the first to show that there are circadian fluctuations in sAA activity, SFR, and pH both before and after acid stimulation. According to *Dawes (1975)*, unstimulated submandibular saliva showed circadian rhythms, significant for the group as a whole saliva, in SFR and in the concentrations of sodium, potassium, magnesium, chloride, and inorganic phosphate. In this study, sAA activity was lower in the morning and then increased gradually. The levels of sAA activity and SFR did not fluctuate significantly among the four time points. Our results are in line with earlier studies from Nater's group (*Nater et al., 2007*; *Rohleder et al., 2004*) and others who took saliva samples two to six times per day. Those studies have shown that there is a reduction of sAA activity at the beginning of the day. However, the results in the afternoon were different from those obtained in the current study, which may reflect the differences in saliva collection and processing. In this study, the collection and processing of saliva were performed by the same person.

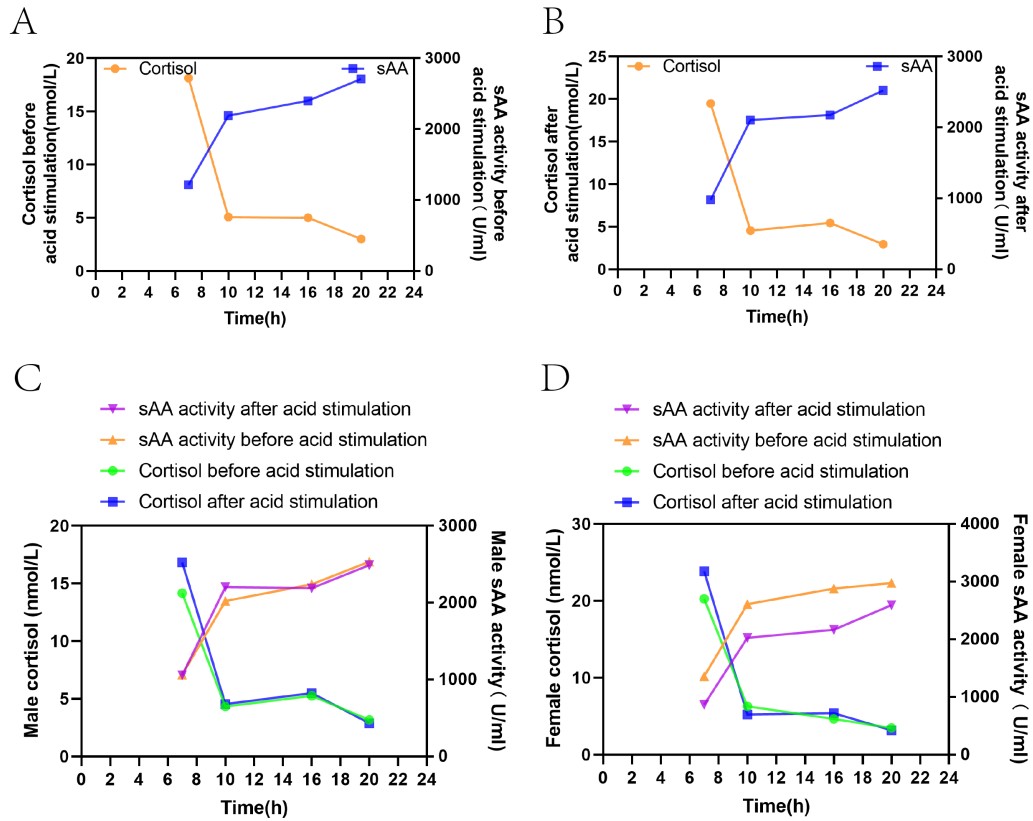

**Figure 3** **Correlation between sAA activity and cortisol before and after acid stimulation.** (A) The change of the two indexes in the resting state. (B) The change of the two indexes after acid stimulation. (C) The change of the two indexes in the resting state and acid stimulation in male volunteers. (D) The change of two indexes in the resting state and acid stimulation in female volunteers.

Salivary cortisol and sAA are the main indicators of the SNS and HPA axis, respectively (*Strahler et al., 2017*). Studies have shown that salivary cortisol and sAA may be potential diagnostic tools for detecting stress-induced heart disease (*Cozma et al., 2017*); salivary cortisol and sAA can be considered important and noninvasive tools for assessing anxiety in children such as for tooth extraction (*Chaturvedi et al., 2018*). If we do not fully understand the diurnal characteristics of salivary cortisol and sAA under resting and acid stimulation states as well as their potential influencing factors, the application value of both biomarkers may be limited. Therefore, a good understanding of the biological information of sAA and cortisol will help improve the application value of these two biomarkers. Previous studies have suggested diurnal variations of salivary amylase and cortisol. There was no significant difference between stimulating and non-stimulating salivary cortisol, suggesting that SFR did not affect salivary cortisol levels (*Brorsson et al., 2014*). This study investigated the fluctuations of salivary sAA and cortisol in healthy young men and women before and after acid stimulation among four time points within one day, while controlling for other possible influencing factors.

The cortisol circadian fluctuations changes before and after acid stimulation, suggesting that cortisol reflects saliva stability index of the circadian rhythm, which is only slightly affected by acid stimulation. The salivary alpha-amylase activity negatively correlated with cortisol level. In the study, sAA activity gradually decreased and cortisol gradually increased during the day, both before and after acid stimulation, and there was a negative correlation between the two, suggesting that salivary cortisol level may predict salivary α-amylase level during the same period.

Salivary secretion is also influenced by gender, and many hormonal systems show gender differences in basal activity or stress response patterns, such as the HPA axis (*Da Mata et al., 2020*). In previous studies, women and men have participated in most experiments to assess the circadian rhythm of sAA and its acute response to stress, exercise, or other conditions. However, few studies have reported circadian changes of salivary secretion in men and women (*Rantonen & Meurman, 2000*). Of 76 participants in a previous study that assessed circadian rhythms, 44 were premenopausal women, and growth curve analysis showed no gender differences in mean sAA concentrations and the slope of circadian rhythms (*Nater et al., 2007*). Similar results were found in a pilot study involving 12 women and five men (*Rohleder et al., 2004*). It has been found that in each stage of the stress response of the public speaking task (GPST-A) in adolescent groups, the cortisol concentration of males was significantly higher than that of females, but there was no gender difference in sAA response (*Katz & Peckins, 2017*). The study found that stress system activity early in life was associated with long-term neurodevelopment, and that gender was an important factor in this relationship: salivary alpha-amylase activity positively correlated with social-emotional development only in boys, and cortisol levels negatively correlated with social-emotional development in boys (*Andiarena et al., 2017*). However, in all these studies, the number of participants was too small and insufficient to rule out potential confounding factors.

This study showed that there were no significant gender differences in sAA activity and cortisol between men and women before and after acid stimulation. One study showed that there were significant differences in the physiological response of men and women to exercise stress, especially in the activation of the SAM and HPA axes represented by α-amylase and cortisol, respectively (*Rutherfurd-Markwick et al., 2017*). there was a weak negative correlation between α -amylase and cortisol in both men and women at rest, but the correlation coefficient increased in men during exercise. Therefore, exercise stimulation may affect the secretion of α-amylase and cortisol. Therefore, there were higher reflexes of sAA and cortisol for acid stimulation or exercise conditions in men compared with women.

## CONCLUSIONS

As an easy to collect and noninvasive examination specimen, saliva has potential application value in the diagnosis and treatment of diseases. This study provided the diurnal changes of sAA, cortisol, pH, and SFR in young men and women before and after citric acid stimulation. sAA and cortisol showed diurnal fluctuation, and the variation characteristics of male and female under resting state and acid stimulation were basically the same,

indicating that the diurnal rhythm of the two indexes were not affected by gender and acid load. Salivary cortisol was shown to be a stable index reflecting the circadian rhythm of salivary secretion. The variation trend of salivary alpha-amylase activity was opposite to that of cortisol, with significant negative correlation. These results may enable the selection of the correct sampling time for research and the selection of appropriate sampling strategies in studies investigating chronic psychosocial conditions.

## ACKNOWLEDGEMENTS

The authors thank Dr. Pengfei Li for his proofreading of the manuscript and the recommendation of the journal. We thank all of volunteers for their participation. We are grateful to everyone who helped with in this work.

### Funding

This work was supported by the Nature Science Foundation of China (No.81973723) and the Natural Science Foundation of Guangdong(No.2021A515011476). The funders had no role in study design, data collection and analysis, decision to publish, or preparation of the manuscript.

### Grant Disclosures

The following grant information was disclosed by the authors:
The Nature Science Foundation of China: No.81973723.
The Natural Science Foundation of Guangdong: No.2021A515011476.

### Competing Interests

The authors declare there are no competing interests.

### Author Contributions

- Haimei Xie conceived and designed the experiments, performed the experiments, analyzed the data, prepared figures and/or tables, authored or reviewed drafts of the paper, and approved the final draft.
- Xiaomei Zheng performed the experiments, prepared figures and/or tables, authored or reviewed drafts of the paper, and approved the final draft.
- Ye Huang, Weihao Li, Wenkai Wang, Qiao Li, Jiangtao Hou, Lulu Luo and Xiuying Kuang performed the experiments, authored or reviewed drafts of the paper, and approved the final draft.
- Chuan-quan Lin conceived and designed the experiments, prepared figures and/or tables, authored or reviewed drafts of the paper, and approved the final draft.

### Human Ethics

The following information was supplied relating to ethical approvals (i.e., approving body and any reference numbers):

The Academic Ethics Committee of The First Affiliated Hospital of Guangzhou University of Chinese Medicine approval to carry out the study within its facilities (Ethical Application Ref:No.K[2019]151.

## Data Availability

The raw measurements are available in the Supplementary File.

## Supplemental Information

Supplemental information for this article can be found online at http://dx.doi.org/10.7717/peerj.13178#supplemental-information.

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
