# Peer review of "Diurnal pattern of salivary alpha-amylase and cortisol under citric acid stimulation in young adults"

_PeerJ, doi:10.7717/peerj.13178_

## Round 0.1 · original submission · Major Revisions

As you will see from the comments, reviewer-2 has identified a significant number of major concerns with the study. My initial inclination was to reject the paper given how extensive this list is, but I have decided to offer you the opportunity to address/rebut these comments. However, I must stress that any rebuttal or revision must address all comments adequately and may still fall short of the decision to accept if in the opinion of myself and external reviewers the alterations are not deemed appropriate.

Some re-analysis of the data will be required.

Reviewer 1 ·

Basic reporting

Xie and colleagues have conducted an essentially sound study on diurnal variances and sex differences in alpha-amylase and cortisol in saliva (as well as flow rate and pH) of young adults. The manuscript is relatively well-written (except for a few lines in the Discussion section, and the first part of the Conclusion) and essentially transparent. I suggest some corrections/clarifications, mainly in the data presentation/analysis and communication of details of experiments/study design.

Lines 240-244 needs revision – the sentence is not readable in the present version. This also applies to Lines 267-270.

Line 307 – what is meant by the statement that the two analytes in men “have higher stress capacities”? It does not make any sense to me.

Lines 310-317 are not typical features of a Conclusion section and could to my opinion be revised/shortened/removed.

Experimental design

Line 89-99: Several (most relevant!) exclusion criteria were used. Were these solely participant-reported, or objectively evaluated by the investigators? The latter would add value to the manuscript.

Line 115-133: Regarding the saliva analyses which were not performed immediately: it would be critical to know how saliva was handled/stored awaiting analysis of sAA and cortisol – fresh (under stabile temperature?), frozen, centrifuged post thawing, etc etc. Also communicate time to analysis and if analytes were measured on several occasions or not.

Line 135-141: Statistics. The authors demonstrate that the data is not normally distributed, and thus correctly evaluate by non-parametric tests. Still, the authors present the results as means and standard errors. More relevant non-parametric descriptive statistics would rather be medians and interquartile ranges, or similar.

Validity of the findings

Figure legend 1 (and onwards). Define abbreviations also in legends – allow figures to be fully readable as standalone parts.

Fig 1-2, 4, 5-6: authors state that comparisons have been made between two groups. Still, stars and bars indicate comparisons between 3 time points. Were time points merged? This must be crystal clear.

Additional comments

Line 195: sAA (typo)

Line 198: significant (typo)

Line 332 (?): within (typo)

Figure 8: cortisol (same typo ocurring in all subpanels)

Reviewer 2 ·

Basic reporting

Overview

This study examined how diurnal levels of salivary alpha amylase (sAA), salivary cortisol, salivary flow rate, and salivary pH levels would be impacted by a citric acid stimulation test. The study was conducted in a sample of healthy male and female participants. Saliva samples were collected at 4 time-points over the course of the day, to map the circadian fluctuations of the salivary markers.
The study found that citric acid stimulation was associated with increased sAA and cortisol levels in men, at certain time points, compared to women, and that neither SFR nor the pH levels were impacted by the citric acid stimulation.
The authors conclude that sAA and cortisol seem to be robust against the citric acid influences and propose that the results provide a comprehensive understanding of the physiological characteristics of human salivary secretion.

I commend the authors for this endeavor, knowing first-hand the challenges that are associated with collecting salivary data at multiple time points during the day.
However, my enthusiasm for this manuscript was overshadowed by the numerous issues with it. Highlighted below are my major issues with the manuscript in its current form.

Abstract:
1. The authors need to make the rationale behind the use of citric acid stimulation more explicit, in both the abstract and the main manuscript.
2. The authors do not need to provide the statistical results of the correlations in the abstract. These should be reserved for the results section in the manuscript.
3. The authors do not make it clear what their findings mean, and how they fit in the literature more broadly.
4. I further find it difficult to understand how the findings from this study “contribute to a more comprehensive understanding of the physiological characteristics of human salivary secretion, and the selection of appropriate sampling strategies for the right salivary sampling time in studies investigating chronic psychosocial conditions.”

Introduction

Overall, I found that the introduction lacked structure, and did not adequately highlight the aim of the study, and the knowledge gaps that it is trying to fill.

1. Once again, the rationale for the study is missing. Specifically what is the purpose behind the use of citric acid stimulation, and what information the effects of this manipulation provides on the diurnal fluctuations of the salivary markers under study.

2. Given that the study was not conducted on clinical or sub-clinical samples, the mention of patient groups in line 70 gives the reader the impression that the study will either include patients or that the findings will have some relevance to patient samples – which does not seem to be the case here.


3. The authors mention that salivary secretion and biomarkers are influenced by sex hormones, specifically mentioning the effects of menstrual cycle on fluctuating levels of sAA. This suggests that the authors accounted for gonadal hormones, and/or menstrual cycle study of their female participants. I cannot find any information confirming this however. Is this the case?

4. The last sentence of the introduction states the goal of the study which is to “identify the diurnal patterns in sAA, SFR, pH, and salivary cortisol before and after citric acid stimulation”. Here the authors introduce SFR and pH as variables of interest, without providing any background literature on why these variables should be considered within the context of diurnal rhythms, and why citric acid stimulation should impact these markers.


5. The study hypotheses are missing. Each hypothesis should be explicit and clear, so that the reader has a clear idea of what the authors aim to investigate and expect.

Experimental design

1. The authors need to provide more information about the participants. Specifically, were the female participants using chemical contraceptives? Or if they were tested in a specific menstrual cycle phase? Or whether these factors were considered or accounted for.

2. It would be helpful to know more about the data collection procedure. Did data collection occur in the lab or in participants’ homes? If in the lab, did the participants come for a full day in the lab? What did they do in the lab between sample collections. More information about the study protocol will be helpful.


3. What was the time delay between the saliva samples that were collected before and after citric acid stimulation?

4. I am still unclear if citric acid stimulation is used as a means to induce stress (line 133)? If so, this needs to be make explicitly clear throughout the manuscript.

Validity of the findings

Statistical analyses

1. Why did the authors choose not to transform the non-normally distributed data? Given the study design, the authors could have run repeated measures anovas or HLMs to investigate the effects of the citric acid stimulation on the salivary markers.


2. The results section does not provide any statistics for the analyses presented.

3. Looking at the sAA values presented, the numbers don’t square up with the values typically seen in the literature. Could the authors provide a reference for their analysis protocol, as well as information about the detection limits of the assays they used.


4. Once again, I feel that a clear goal of the study with explicit hypotheses will help streamline the results section. Is there a reason that the authors conducted the analyses first in the whole sample, and then spilt the groups by sex? Why not include sex as a between subjects factor in the analyses from the outset?

5. Figures 5 and 6 can be combined together to visually represent the changes in the salivary markers over time in men and women.


6. I am not sure what information figures 4 and 7 (the ratio of the markers) are depicting, to help the reader interpret the results.

7. It would be helpful if the pre-and post- stimulation cortisol and sAA data in figure 8 were plotted in the same graph.


Discussion
1. In lines 234-240 the authors’ discussion of the clinical applications of sAA and cortisol do not fit with the scope of the paragraph, and the study in general, and appear to be contextually irrelevant.

2. In lines 254-262, the authors repeat the results of the study but without putting them in the broader context of the literature, nor do they provide any interpretations for what these findings mean.


3. Lines 259-262, the authors appear to over extend the relevance of their findings. How do these results “…..contribute to a more comprehensive understanding of the physiological characteristics of human salivary secretion, as well as to the selection of appropriate sampling strategies for the right salivary sampling time in studies investigating chronic psychosocial conditions”? It is recommended that the authors expand on these claims, or perhaps temper their statement.

---

## Round 0.2 · accepted · Accept

Thank you for your detailed rebuttal and revised manuscript. The reviewer and I agree that this is now acceptable for publication.

Reviewer 1 ·

Basic reporting

no comment

Experimental design

no comment

Validity of the findings

no comment

Additional comments

no comment